# REGULATORY FOCUS: PROMOTION AND PREVENTION INCLINATIONS IN POLICY SEARCH

## ABSTRACT

The estimation of advantage is crucial for a number of reinforcement learning algorithms, as it directly influences the choices of future paths. In this work, we propose a family of estimates based on the order statistics over the path ensemble, which allows one to flexibly drive the learning process in a promotion focus or prevention focus. On top of this formulation, we systematically study the impacts of different regulatory focuses. Our findings reveal that regulatory focus, when chosen appropriately, can result in significant benefits. In particular, for the environments with sparse rewards, promotion focus would lead to more efficient exploration of the policy space; while for those where individual actions can have critical impacts, prevention focus is preferable. On various benchmarks, including MuJoCo continuous control, Terrain locomotion, Atari games, and sparse-reward environments, the proposed schemes consistently demonstrate improvement over mainstream methods, not only accelerating the learning process but also obtaining substantial performance gains.

## 1 INTRODUCTION

It is not enough to do good; one must do it the right way.

The best-known motivational principle that has been recognized in psychology (Gray, 1982; Atkinson, 1964) is the hedonic principle that people approach pleasure and avoid pain. However the principle does not explain *how* people approach and avoid. Higgins (1997) proposes the regulatory focus theory to address this question. A promotion focus is concerned with advancement, growth, and accomplishment, and a prevention focus is concerned with security, safety, and responsibility (Crowe & Higgins, 1997). The theory hypothesized that the promotion focus inclination insures against errors of omission, and the prevention focus inclination insures against errors of commission. Furthermore, the regulatory fit theory (Higgins, 2000; Higgins et al., 2003) proposes that when people pursue a goal in a manner that fit their regulatory orientation, they are stronger motivated during goal pursuit. For example, Lee & Aaker (2004) showed that ads are more persuasive if there is a regulatory fit: use gain frames (*e.g.* "great looks and exceptional engineering") for individuals who are eager to approach positive outcomes, and use loss frames (*e.g.* "don't be stranded with a disabled vehicle without an emergency safety kit") for individuals who are vigilant to avoid negative outcomes.

We study the phenomenon of regulatory fit in the context of reinforcement learning (RL). Be aware that humans own real emotions and are fundamentally more complex than RL agents, thus we borrow terminologies in psychology while trying to retain the resemblance. We hypothesize that different RL environments are more fit with agents of different regulatory focus. For example, in the environments with sparse rewards, it might be advisable to actively exploring those policies that show potentials without being discouraged by a few failed trials; while in the environments that are fragile, *e.g.* those where a wrong choice of action can lead to catastrophic consequences down the path, a more conservative, risk-averse stance might be preferable. There are three questions need to answer: (1) How to automatically determine the suitable regulatory focus for an environment; (2) How to train an agent with a specified regulatory focus; (3) Is there any benefit when training the agent with the regulatory focus that fits the environment? In this article, we aim to answer the second and third questions and leave the first question open.

The research on deep reinforcement learning is gaining momentum in recent years. A number of learning methods, such as Proximal Policy Optimization algorithm (PPO) (Schulman et al., 2017),

Trust-Region Policy Optimization algorithm (TRPO) (Schulman et al., 2015a), and Advantage Actor-Critic algorithm (A2C) (Mnih et al., 2016), have been developed. These methods and their variants have achieved great success in challenging problems, *e.g.* continuous control (Schulman et al., 2015b), locomotion (Heess et al., 2017), and video games (Vinyals et al., 2017). The core of all these methods is the estimation of the advantage $A(s_t, a_t)$, *i.e.* the gain in the expected cumulative reward relative to the state value $V(s_t)$ if a certain action $a_t$ is taken. A common practice is to use the $n$-step estimate over the sampled trajectories. This way has been widely used in actor-critic algorithms such as the A2C. Schulman *et al.* presents a generalization called Generalized Advantage Estimator (GAE) (Schulman et al., 2015b), which blends 1 to $n$-step estimators with exponentially decayed weights, thus reducing the variance of policy gradients. For convenience, we refer to the set of 1 to $n$-step estimators along a trajectory as the *path ensemble*.

We propose to take the maximum over path ensemble to implement the promotion focus inclination, and take the minimum over path ensemble to implement to prevention inclination during training. Intuitively, when taking the maximum or minimum, the agent is comparing a list of counterfactual speculations. When it looks at a $k$-step estimation where $k \leq n$, it is thinking what if it didn't took those actions after step $k$? For a promotion focus, in order to insure hits and insure against omission, it would favor the estimation with largest value because it has the possibility of being that good. Similarly, for a prevention focus, it prefers the minimum to insure against errors of commission.

We systematically studied different estimation schemes on various environments, including the sparse-reward environments (Fu et al., 2017), the Terrain RL Simulator (Berseth et al., 2018) for biped locomotion, the Atari games (Bellemare et al., 2013), and the set of MuJoCo environments for continuous control (Brockman et al., 2016). Our study shows that the proposed estimation strategies, when chosen appropriately, can substantially outperform the standard way. In particular, the promotion focus inclination, *i.e.* taking the maximum over path ensemble, greatly improves the learning efficiency in the environments with sparse rewards. On the other hand, the prevention focus inclination, *i.e.* taking the minimum, can effectively stabilize the learning by avoiding risky actions.

It is noteworthy that the proposed strategic inclinations differ essentially from the recent two lines of work on robustness (Smirnova et al., 2019; Delage & Mannor, 2007) and risk-sensitivity (Tamar et al., 2012; Chow & Ghavamzadeh, 2014). Robust MDP optimizes for the worst case when there exist uncertainties in the parameters; risk-sensitive MDP optimizes the value of a risk measure. Our method does not modify the training objective. Instead, it controls the policy gradient towards policies with promotion or prevention inclinations through the alternative ways of estimating the advantage.

## 2 RELATED WORK

**Estimation of Advantage and Action-Value**   The $n$-step advantage estimation is derived from the $n$-step return, which is used in the $n$-step TD or the $n$-step Sarsa method (Sutton & Barto, 2011). Similarly, the generalized advantage estimator (Schulman et al., 2015b) is analogous to the $\lambda$-return (Watkins, 1989) in the TD($\lambda$) method (Seijen & Sutton, 2014). They are all linear combinations over the path ensemble. A nonlinear combination scheme is seen in the work of Twin Delayed DDPG (TD3) (Fujimoto et al., 2018), which uses the $\min$ of two critics to estimate $Q$. However, it aims to mitigate the overestimation problem in deep Q-learning, and the objects be combined are the network outputs instead of the estimators based on the cumulative rewards. Another nonlinear combination is the Positive Temporal Difference (PTD) (Van Hasselt & Wiering, 2007; van Hasselt, 2012), where the advantage is set as 1 when the 1-step estimate of advantage is positive, and is set to 0 otherwise. The benefit of limiting policy updates only toward those actions that have positive advantages is increasing the stability of learning.

**Robustness and Risk-Sensitivity in RL**   In the assumption of robust MDPs (Nilim & El Ghaoui, 2004; Xu & Mannor, 2007; Delage & Mannor, 2010; Mannor et al., 2012; Smirnova et al., 2019), the parameters of the problem lie in an uncertainty set, and the target is to find a solution that performs best under the worst cases. On the other hand, risk-sensitive MDPs consider the uncertainty of the rewards. The objective is to minimize the risk-measure, which is defined by the exponential utility (Howard & Matheson, 1972), the Conditional Value-at-Risk (CVaR) (Chow & Ghavamzadeh, 2014; Prashanth, 2014; Chow et al., 2015; Tamar et al., 2015), the percentile (Delage & Mannor, 2007) or the variance (Tamar et al., 2012; Prashanth & Ghavamzadeh, 2013) of the cumulative

rewards. The statistics over the path ensemble is not the same as those statistics for advantage. However, using a suitable nonlinear combination of the numbers in the path ensemble, the exploration strategy in our algorithms can bias the learning toward risk-averse or risk-seeking policies. Unlike the robust and risk-sensitive approaches, which typically introduce involved computations, our approach is straightforward to implement and requires minimal changes to incorporate the idea into existing algorithms such as A2C, PPO, TRPO and their variants.

**Distributional RL** The value distribution is the object of study in distributional RL. Since the full distribution of return is difficult to estimate, researchers have adopted nonparametric methods (Morimura et al., 2010) or used simple distributions (Bellemare et al., 2017; Dabney et al., 2018; Barth-Maron et al., 2018) to approximate it. The RL algorithms can be formulated with any criterion based on the distribution of return, such as the aforementioned risk-sensitivity measures. Again, the distribution formed by elements in the path ensemble does not represent the full distribution of return, but some joint distribution that we can exploit.

## 3 REGULATORY FOCUS BY ORDER STATISTICS OVER PATH ENSEMBLE

In this section, we first review the preliminary knowledge on RL. Then we introduce the central concept of this article: regulatory focus implemented by estimators over path ensembles. Specifically, we focus on the family of order statistics. Next, we give an illustrative study on the $\max$ statistics over the path ensemble, where we show that the induced estimator of action-values would influence the optimization process of RL algorithms. With this specific example in mind, we give a practical algorithm to incorporate the general estimators into existing RL algorithms in the last section.

### 3.1 PRELIMINARY

We consider the standard formulation of RL, which is typically modeled as an MDP $(\mathcal{S}, \mathcal{A}, \mathcal{T}, \gamma, R)$, where $\mathcal{S}$ is the state space, $\mathcal{A}$ is the action space, $\mathcal{T} = \{P_{sa}(\cdot) \mid s \in \mathcal{S}, a \in \mathcal{A}\}$ is the transition probabilities, $\gamma \in (0, 1]$ is the discount factor, and $R$ is the reward function. At timestep $t$, the agent in state $s_t$ interacts with the environment by choosing an action $a_t \sim \pi(s_t)$ following the policy $\pi$, and receives a reward $R_t$ from the environment. The environment then transits to the next state $s_{t+1}$. The discounted return is defined as $G_t := \sum_{i=t}^{T-1} \gamma^{i-t} R_i$, and the goal of RL is to maximize the expected return $J = \mathbb{E}_{s_0 \sim S_0}[G_0 \mid s_0]$, where $S_0$ is the distribution of initial states. The action-value function $Q^\pi(s_t, a_t) := \mathbb{E}[G_t \mid s_t, a_t]$ under a policy $\pi$ is the expected return of taking action $a_t \sim \pi(s_t)$ in state $s_t$. In policy-based model-free deep RL algorithms, policy $\pi_\theta$ with parameters $\theta$ is optimized via gradient ascent. An estimation of the gradient is the policy gradient

$$\nabla_\theta J(\theta) = \mathbb{E}_{\tau \sim \pi_\theta(\tau)}\left[ \sum_{t=0}^{T} \frac{\nabla_\theta \pi_\theta(a_t \mid s_t)}{\pi_\theta(a_t \mid s_t)} A(s_t, a_t) \right], \tag{1}$$

where $\tau = \{s_0, a_0, s_1, a_1, \ldots, s_T, a_T\}$ is a trajectory following the policy. $A(s_t, a_t) := Q(s_t, a_t) - V(s_t)$ is the advantage function. The $n$-step look ahead estimation of advantage function is given by

$$\hat{A}_t^{(n)} := \sum_{i=0}^{n-1} \left( \gamma^i r_{t+i} \right) + \gamma^n V(s_{t+n}) - V(s_t), \quad n = 1, 2, \ldots, T - t. \tag{2}$$

### 3.2 ORDER STATISTICS OVER THE PATH ENSEMBLE

Given a trajectory, the set of $n$-step estimators $\mathcal{E} := \{\hat{A}_t^{(1)}, \hat{A}_t^{(2)}, \ldots\}$ is defined as the *path ensemble* for the pair $(s_t, a_t)$. We have briefly mentioned that the maximum over path ensemble can achieve the promotion focus, and the minimum over path ensemble can achieve the prevention focus in Introduction. The $\max$ statistics and the $\min$ statistics over the path ensemble are respectively computed by

$$\hat{A}_t^{\max} := \max_i \left\{ \hat{A}_t^{(i)} \right\}, \quad \hat{A}_t^{\min} := \min_i \left\{ \hat{A}_t^{(i)} \right\}. \tag{3}$$

For many environments that are neither sparse nor fragile, whether they are more close to promotion focus or prevention focus is unclear without in-depth inspection. They also might be too complex to

be simply described by any one of the two regulatory types. For these cases, we propose a generalized order statistics which is a hybrid of the maximum and minimum. We call it the max-abs statistics which took the element with the maximum absolute value,

$$\hat{A}_t^{\text{max-abs}} := \operatorname*{argmax}_{A \in \left\{ \hat{A}_t^{(i)} \right\}} |A|. \tag{4}$$

To understand how elements in the path ensemble relates to regulatory focus, we interpret the $k$-step estimation as counterfactual speculations when $k \leq n$. Since $k$-step estimation only involves the first $k$ states after the evaluated state, all actions and the followed observations are dropped and be replaced by the state-value at step-$k$. This underlies a counterfactual thinking that what if the agent didn't do the action sequence $a_{t+k+1}, a_{t+k+2}, \ldots$, and replace the situation by the average behavior? Then different regulatory focus would treat the set of counterfactual speculations in different ways.

**The** max **Statistics and the Promotion Focus**   A promotion focus is concerned with advancement, growth, and accomplishment. The promotion focus inclination is to insure hits and insure against errors of omission. Facing the set of counterfactual speculations, it would choose the one with the maximum value to insure any action $a_t$ in $s_t$ that is possibly to behave good to have more importance. The max statistics leads to optimistic estimation of advantages. It is benefical for sparse-reward environments as actions that may lead to large returns will not be buried by the later bad actions.

**The** min **Statistics and the Prevention Focus**   A prevention focus is concerned with security, safety, and responsibility. The prevention focus inclination is to insure correct rejections and insure against errors of commission. Facing the set of counterfactual speculations, it would choose the one with the minimum value to avoid wrongly acknowledge an action that only temporarily leads to high rewards. The min statistics implies the risk-averse bias. It avoids actions that may cause bad states later. By decreasing the advantage estimation of those actions, it makes the optimization direction away from them. This property is useful for fragile environments, such as the biped locomotion environments which are sensitive to joint motions of the characters.

**The** max**-abs Statistics and Overreaction**   In the max-abs statistics, the estimator with the largest absolute value in the path ensemble is chosen. This means that an action is evaluated as either overly good when the largest positive advantage is selected, or overly bad when the smallest negative advantage is selected. We refer to this behavior as *overreaction*. Overreaction is not a terminology in regulatory focus theory, but a descriptive term for the phenomenon concerning the effects of the max-abs statistics. This statistics implements a heuristic that makes the good paths look better and the bad paths look worse. As will be shown in experiments, this heuristic generally improves sample efficiency for the MuJoCo and Atari environments, where their regulatory focuses are unclear.

Both the maximum and the minimum belong to the family of order statistics, where the $k$-th order statistics equals to the $k$-th smallest value. From computational perspective, it is also viable to test whether the more general order statistics are effective for training, although lacking strong explanations. We put the study on general order statics in Appendix A.

### 3.3   AN ILLUSTRATIVE STUDY ON THE max STATISTICS

Before delving into the discussion of estimating advantages, we first look at a concrete example of how the max statistics of $Q$-values affect the learning process of the policy iteration algorithm (Sutton & Barto, 2011, Section 4.3). In Figure 1(a), an MDP with 6 states is drawn. For simplicity, we assume that both the state transitions and the rewards are deterministic, and the discount factor is 1. In this example, only the two actions in state $s_3$ can get rewards, where $R(s_3, a_1, s_5) = -2$ and $R(s_3, a_2, s_6) = 2$. At the initialization step of the policy iteration algorithm, the policy is initialized as the uniform random policy $\pi_1$. In the first step of the policy iteration, we evaluate the $Q$ function and value function $V$ of this random policy $\pi_1$. Under policy $\pi_1$, all possible trajectories are also shown in Figure 1(a). We compute that $Q^{\pi_1}(s_1, a_1) = 0$, and $Q^{\pi_1}(s_1, a_2) = 0$. So in the next step, the greedy policy $\pi_2$ for state $s_1$ is still at random. We compute $\hat{Q}^{\text{max}}$ using the max statistics,

$$\hat{Q}^{\text{max}}(s_1, a_1) = \mathbb{E}\left[ \max_i \{R^{(i)}\} \mid s_1, a_1 \right] = 0, \quad \hat{Q}^{\text{max}}(s_1, a_2) = \mathbb{E}\left[ \max_i \{R^{(i)}\} \mid s_1, a_2 \right] = 1. \tag{5}$$

Figure 1: Toy examples using the $\max$ path ensemble. (a) When calculating return of the second trajectory, the $\max$ chooses $G^{(1)}$, so that the bad action $a_1$ at state $s_3$ is blacklisted. This leads to an optimistic view when evaluating the long-term returns of actions. The dashed circle means that it is ignored in the computation of return. (b) and (c) demonstrate the over-estimation problem of the $\max$ statistics when uncertainty exists in transitions or rewards.

Using $\hat{Q}^{\max}$, the greedy policy at state $s_1$ is $\arg\max_a \hat{Q}^{\max}(s_1, a) = a_2$. Actually, the algorithm with max statistics finds the optimal policy within only 1 step in this example.

We see that the $\max$ statistics over the path ensembles converges faster in the toy example because it discovers the potentially good action $a_2$ at state $s_1$ earlier than using the standard way. As the computation of the second trajectory in Figure 1(a) shows, the $\max$ statistics blacklists the bad action $a_1$ in state $s_3$. In a trajectory, when an action is so bad that the later actions cannot compensate for the caused loss, it will be replaced by the average actions. This good-action discovering ability is a characteristic of promotion focus, and is suitable for sparse-reward environments. In the early exploration stage in the training, the agent barely receives positive signals. Many trajectories might happens to act well at some time, but then go to bad states later due to wrong actions. The $\max$ statistics can highlight actions that are possible to get high rewards in any middle step of the trajectory. By effectively discovering good actions, this method is expected to improve the sampling efficiency.

The foundational position of advantage in actor-critic algorithms is attributed by the policy gradient theorem in Equation (1). A sample $(s_t, a_t)$ makes the policy network $\pi_\theta$ adjust its parameters according to the estimation of advantage $\hat{A}_t(s_t, a_t)$. The probability of $\pi_\theta(s_t) = a_t$ is increased when the advantage is positive, and decreased when the advantage is negative. If the estimation of advantage $\hat{A}_t(s_t, a_t)$ is manipulated to be larger, the learned policy is then biased toward the action $a_t$ at state $s_t$. Thus when the estimation of certain advantages $\hat{A}_t(s_t, a_t)$ is manipulated properly, the learned policy can be biased toward the desired regulatory orientation.

## 3.4 INCORPORATING REGULATORY FOCUS INTO RL ALGORITHMS

We should be careful that in the presence of uncertainty or noise, applying the $\max$ statistics in policy iteration may fail to improve policy. We give two examples when the estimation is overly optimistic. The first example, which is shown in Figure 1(b), is caused by the uncertainty in state transitions. We have $\hat{Q}^{\max}(s_1, a_1) = 0.5$ and $\hat{Q}^{\max}(s_1, a_2) = 1$, which implies that action $a_2$ should be chosen in state $s_1$. However, the optimal strategy is to select $a_1$. This is caused by the the $\max$ statistics' ignorance on the coupling risks. When taking an action, both good and bad may happen. The bad cases is ignored in the estimation. However, we cannot avoid bad next-state when selecting that action. The second example, shown in Figure 1(c), is a case when the reward owns randomness. The problem is that the maximum operation over the ensemble causes overestimation, a symptom that also troubles the $Q$-learning algorithm as discussed in TD3 (Fujimoto et al., 2018).

We overcome this shortcoming from two aspects: (1) implement the regulatory focus as an *exploration* strategy. Namely, the estimation from path ensemble is used with a probability $\rho$. We call $\rho$ as the *regulatory ratio*. (2) using a small ensemble size. We argue that a small ensemble size is sufficient for practical use because elements in the path ensemble are inter-related. Roughly speaking, the difference between the $i$-step advantage estimator $\hat{A}_t^{(i)}$ and the $j$-step estimator $\hat{A}_t^{(j)}$ in a path ensemble is discounted exponentially with the minimum step index $\min(i, j)$ of the two. In fact, let $j > i$, then

$$\hat{A}_t^{(j)} - \hat{A}_t^{(i)} = \gamma^i \left( \sum_{s=0}^{j-i-1} (\gamma^s r_{t+s+i}) + \gamma^{j-i} V(s_{t+j}) - V(s_{t+i}) \right). \tag{6}$$

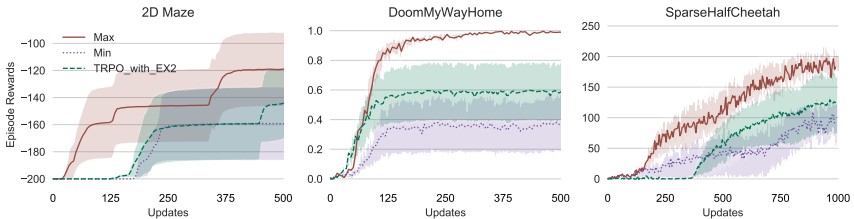

Figure 2: The max and the min statistics on three different types of sparse-reward environments. The max statistics were significantly better than the original EX2 algorithm on all three environments. The min statistics were not effective for sparse reward tasks. All settings were run for 5 seeds.

The sum in the big bracket can be assumed to be bounded. Based on this observation, the values of the estimation $\hat{A}_t^{(k)}$ would not change too much when $k$ becomes large. In practice using a subset of the full ensemble also reduces the computational cost. For example, we used an ensemble with only four elements $\mathcal{E} = \{\hat{A}_t^{(1)}, \hat{A}_t^{(16)}, \hat{A}_t^{(64)}, \hat{A}_t^{(2048)}\}$ in the MuJoCo physics simulator experiments.

Summarizing the above discussion, our algorithm is designed as follows.

---

**Algorithm 1** Regulatory Focus by Order Statistics over Path Ensembles

---

1: Parameters: the ensemble set $\mathcal{E}$, the order statistics $\hat{A}^{\text{order}}$ over $\mathcal{E}$, and the regulatory ratio $\rho$.
2: **for** each iteration **do**
3:     Collect trajectories.
4:     Compute the advantage $\hat{A}$ and the estimation $\hat{A}^{\text{order}}$.
5:     Update the policy network according to PPO, A2C, or TRPO, in which $\hat{A}^{\text{order}}$ is used with probability $\rho$, and $\hat{A}$ is used otherwise.
6:     Update the critic network using the estimation of $Q$ value, which equals to $V + \hat{A}$.
7: **end for**

---

Our method can be easily plugged into any actor-critic algorithms with a minimal code change and a small computational overhead.

## 4 EXPERIMENT

We evaluate the performance of our algorithm on four different types of problems, including the environment with sparse reward, biped locomotion, continuous control and Atari games.

### 4.1 SPARSE REWARD AND THE PROMOTION FOCUS INCLINATION

The amortized EX2 algorithm (Fu et al., 2017) is designed for tasks with sparse rewards. Three environments were chosen from their paper, which represent three types of problems. The order statistics over path ensemble was implemented by modifying the advantage estimation procedure of the authors' original code, where the TRPO is used for policy optimization. The path ensemble was composed of $k$-step estimators where $k \in \{1, 16, 64, 4000\}$ for Maze and Doom, and $k \in \{1, 16, 64, 5000\}$ for SparseHalfCheetah. The numbers 4000 and 5000 originated from the respective batch sizes. The regulatory ratio for both the max and the min statistics was set to $\rho = 0.5$ for Maze and Doom, and $\rho = 0.3$ for SparseHalfCheetah. All hyper-parameters followed the settings of the amortized EX2. The result is shown in Figure 2.

**2D Maze** This environment provides a sparse reward function, where the agent can only get the reward when it is within a small radius of the goal. From the figure, we observed that the max statistics started to gain rewards at the very early stage, which indicates the promotion focus provides a supreme sample efficiency over other methods. After 500 updates, the average episodic reward of max was much higher than that of the EX2 algorithm. The min statistics was worse than EX2.

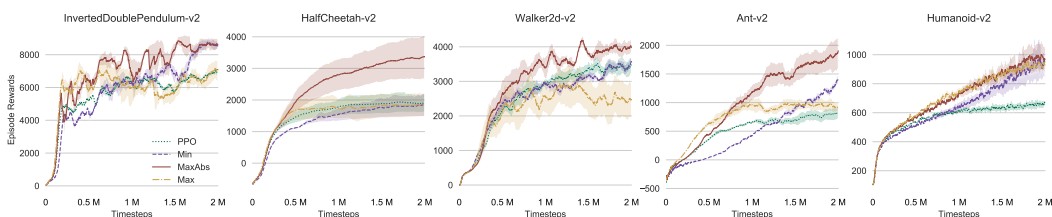

Figure 3: Left: screenshot of the flat and incline environment for biped walking. Right: the performance of PPO and PPO with min and max statistics. We ran 5 seeds for each setting.

Figure 4: Experiments on MuJoCo environments. Three order statistics max, min and max-abs were compared with the vanilla PPO algorithm. Each setting was run with 5 seeds.

**Doom**   MyWayHome is a vision-based maze navigation benchmark, where the agent is required to navigate through a series of interconnected rooms before reaching the goal. A +1 reward is given for reaching the goal before timeout. The task is challenging because the inputs are realistic images in the first-person perspective. The max statistics was very effective for this challenge and it reached nearly 100% success rate. In comparison, EX2 only got an average of 0.6 episode rewards.

**SparseHalfCheetah**   It is a challenging continuous control task with sparse reward. Our promotion focus significantly improved the sample efficiency, and the reward after 1000 updates was 40% higher than the EX2 algorithm. Also, the min statistics was less effective than EX2 in all three benchmarks, which demonstrates that prevention focus is not effective in sparse environments.

## 4.2 BIPED LOCOMOTION AND THE PREVENTION FOCUS INCLINATION

The prevention focus is very useful for environments that are fragile to actions, *i.e.*, one wrong action would lead to catastrophic results later on. The Terrain RL simulator (Berseth et al., 2018) provides such environments. Two biped walking tasks were selected, where a character needs to learn how to walk steadily forward in two different terrains, the flat and the incline, as shown in Figure 3. The character continuously receives rewards until it fells down or is stuck, in which case the episode terminates. This task is challenging because the observation only contains the pose of the character, which forces the character to learn how to walk without memorizing actions based on its location. In contrast, the locomotion tasks in MuJoCo environments contain absolute world coordinates in the observation. The action space is 11-dimensional, which corresponds to the joints of the character. The environment is fragile. If a wrong action is performed, the character might lose balance and fall down. We used PPO in this experiment. The hyper-parameters were borrowed from those designed for MuJoCo environments in the baselines' (Dhariwal et al., 2017) PPO implementation and are also listed in Appendix. The min and max statistics were implemented on top of the PPO algorithm. The path ensemble consisted of $k$-step estimators where $k \in \{1, 16, 64, 2048\}$, and the regulatory ratio is $\rho = 0.3$. The results are shown in Figure 3. The prevention focus inclination via the min statistics successfully mastered the task while the vanilla PPO algorithm failed. The promotion focus inclination by the max was not very effective in these environments as the low reward curves indicate that the character fells down at the beginning of the episode. We conclude that prevention focus by the min statistics of the path ensemble helps the agent to learn in fragile environments.

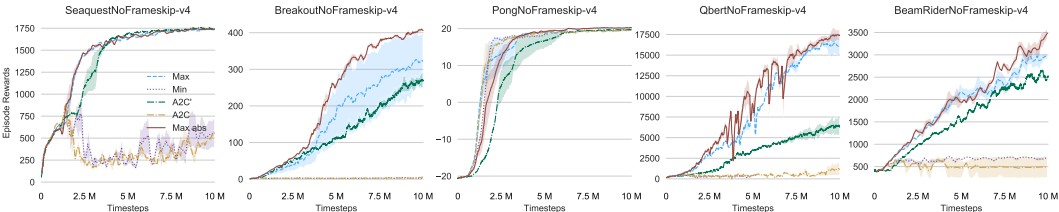

Figure 5: Experiments on the Atari environments. A2C' was the original A2C algorithm without mini-batch, and A2C was using the same mini-batch configuration as other three settings: $\max$, $\min$ and $\max$-abs. Each setting was run for 3 seeds. (This figure is best viewed in color.)

### 4.3 CONTINUOUS CONTROL AND THE OVERREACTION INCLINATION

We tested various advantage estimations on 5 continuous control benchmarks based on the MuJoCo physics simulator. They are not sparse-reward environments, and most of them are also not sensitive to individual actions. For example, the design of the HalfCheetah agent makes it seldom fall down or be stuck. We used PPO in this set of experiments. The implementation was based on the baselines (Dhariwal et al., 2017) code repository, and the default hyper-parameters were adopted. We tested three order statistics including the $\max$, the $\min$ and the $\max$-abs statistics. In all settings, the path ensemble had index set $\{1, 16, 64, 2048\}$, and the regulatory ratio was set to $\rho = 0.4$. Results are shown in Figure 4. We observed that the $\max$-abs statistics was consistently better than the baseline PPO algorithm, while the performances of the $\max$ and the $\min$ statistics depended upon the specific environment. So we conclude that the overreaction inclination by the $\max$-abs statistics is generally effective for a wide range of environments.

### 4.4 ATARI GAMES AND THE OVERREACTION INCLINATION

The new advantage estimations were also tested on a subset of Atari game environments. We used the A2C algorithm with 4 paralleled threads. The implementation was based on the A2C codes in baselines (Dhariwal et al., 2017). Hyper-parameters were the defaults for Atari games. Since the sampled trajectory length was 5 in the default setting, the path ensemble had fewer elements, which was composed of $\{1, 3, 5\}$. The maximum trajectory length of 5 in a batch was too small to gather enough data on $k$-step estimations when $k > 1$. For example, only the first state has a valid 5-step advantage estimation in a length-5 trajectory. This affects the power of the path ensemble. To circumvent this limitation, we collected paths of length $5n$, and then computed the advantage estimators using these longer trajectories. Since the batch size was $n$ times of the original setting, it was split into $n$ mini-batches. For Seaquest and Breakout, $n = 20$; for Pong and Qbert, $n = 10$; and for BeamRider, $n = 60$. The result is shown in Figure 5. In these environments, the sparsity and fragility were unknown. Obviously, the overreaction inclination generally improves the performance, whereas the $\max$ and the $\min$ statistics are only effective for a subset of environments.

## 5 CONCLUSION

According to the regulatory fit theory in psychology, when people engage in goal pursuit activities in a manner that fits their regulatory orientation, they feel right about what they are doing. In this paper, we proposed a simple yet effective way of exploration using the idea of regulatory fit. The regulatory focus is implemented in the context of reinforcement learning via the order statistics over path ensembles, which formed nonlinear combinations of different $n$-step advantage estimators. The maximum, the minimum, and the element with the maximum absolute value were studied in detail. We incorporated these advantage estimators into three widely used actor-critic algorithms including A2C, TRPO and PPO. When the promotion or prevention focus inclination fits the environment, the proposed algorithm could be effective in solving sparse-reward environments, or the fragile environments which are sensitive to individual actions. We verified the effectiveness of our approach by extensive experiments on various domains, including the continuous control, locomotion, video games, and sparse-reward environments.

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

Figure 6: The order statistics over the path ensemble in MuJoCo environments. $x$-axis labels the different order statistics. The four series in the sub-figures were varying the regulatory ratio $\rho$, and the flat line indicated performance of the baseline PPO algorithm. Each data point was averaged over 5 runs.

Hado Van Hasselt and Marco A Wiering. Reinforcement learning in continuous action spaces. In *2007 IEEE International Symposium on Approximate Dynamic Programming and Reinforcement Learning*, pp. 272–279. IEEE, 2007.

Oriol Vinyals, Timo Ewalds, Sergey Bartunov, Petko Georgiev, Alexander Sasha Vezhnevets, Michelle Yeo, Alireza Makhzani, Heinrich Küttler, John Agapiou, Julian Schrittwieser, et al. Starcraft ii: A new challenge for reinforcement learning. *arXiv preprint arXiv:1708.04782*, 2017.

Christopher John Cornish Hellaby Watkins. Learning from delayed rewards. 1989.

Huan Xu and Shie Mannor. The robustness-performance tradeoff in markov decision processes. In *Advances in Neural Information Processing Systems*, pp. 1537–1544, 2007.

## A    ABLATION STUDY

In this section we study the behavior of general order statistics on MuJoCo environments, and then discuss the sensitivity of the algorithms to hyper-parameters.

**The General Order Statistics**    As shown in Figure 6, in three (InvertedDoublePendulum, Ant, and Humanoid) of the five environments, the max and min were better than the 2nd and the 3rd order statistics. This means that some environments welcome both optimistic and risk-averse exploration. Overall, the intermediate order statistics was better than the baseline, which used the GAE estimator.

**Sensitivity on the Regulatory Ratio**    Generally speaking, the regulatory ratio $\rho$ had a great influence on the performance. As shown in Figure 6, the range $[0.2, 0.4]$ is a plausible choice for most experiments. If the ratio is further increased, the performance may degrade. This is consistent with our analysis, and shows the necessity of using regulatory focus as an exploration strategy.

**Effect of Ensemble Size**    We built another path ensemble whose index set consisted of 12 elements. They were $\{1, 2, 4, 8, 16, 32, 64, 128, 256, 512, 1024, 2048\}$. Final performances of the three order statistics, max, min and max-abs, were tested under the regulatory ratio $\rho = 0.4$. For the ensemble with 4 elements $\{1, 16, 64, 2048\}$, the average episode reward in the end of the training was $928, 927, 974$, respectively; and for the ensemble of size 12, the numbers were $767, 854$ and $867$. It shows that a small ensemble size can already get most of the performance boost.

## B    COMPARE WITH $\epsilon$-GREEDY EXPLORATION

As an exploration method, it is helpful to compare the regulatory focus exploration with the basic $\epsilon$-greedy exploration. The results are shown on Mujoco environments in Figure 7. We can see that completely random exploration, even in small proportion, makes the results worse.

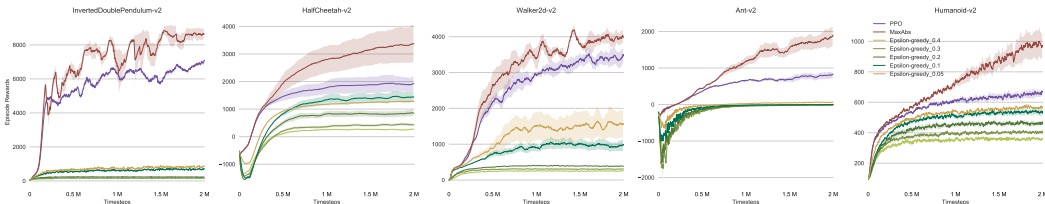

Figure 7: **The $\epsilon$-greedy exploration.** Red: MaxAbs, Purple: PPO, Rest: $\epsilon$-greedy explorations, where $\epsilon$ values of $0.05, 0.1, 0.2, 0.3, 0.4$. Each experiment has $5$ runs.

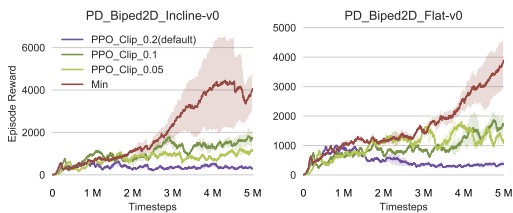

Figure 8: **Conservative policy via clipping.** Red: the Min path-ensemble with the default clipping parameter 0.2, Purple: PPO with the default clipping parameter 0.2, Rest: PPO with smaller clipping constants 0.1 and 0.05. Each experiment has $5$ runs.

## C    COMPARE WITH DIFFERENT CLIP THRESHOLD OF PPO

The clipping threshold of PPO also affects the estimation of advantage. A smaller clipping threshold would result in a more conservative policy iteration. We compare the effect of smaller clipping threshold with the min statistics over path ensemble on the biped locomotion task. Results are shown in Figure 8. Clipping is adding a global minimum constraint, meaning that all trajectories share the same clipping constant. In contrast, the min of path ensemble gives prevention focus inclination in a momentary manner, *i.e.* per trajectory. The performance of min path ensemble is much better.

## D    PATH ENSEMBLE OF GAEs

The generalized advantage estimator $\text{GAE}(\gamma, \lambda)$ is an exponentially-weighted average of different $n$-step estimators,

$$\text{GAE}(\gamma, \lambda) := (1 - \lambda) \left( \hat{A}_t^{(1)} + \lambda \hat{A}_t^{(2)} + \lambda^2 \hat{A}_t^{(3)} + \cdots \right). \tag{7}$$

With different hyper-parameter $\lambda$, the estimator focus on different time ranges. For example, if $\lambda$ is small, then the decay would be very fast and effectively only the first steps in the trajectory is considered; if $\lambda$ is close to 1, then GAE takes longer time ranges into account. Having the above observation, we can use the path ensemble of GAEs with different $\lambda$s as an alternate version of the path ensemble of $n$-step advantages. The GAE formulation would have a smaller variance and may be beneficial for some environments. We tested the formulation of ensembling GAEs on the Mujoco environments and found substantial improvements on the Ant-v2 and Humanoid-v2 environments. Note also that on HalfCheetah-v2 and Walker2d-v2, performance drops compared to ensembling $n$-step advantages. The results are drawn in Figure 9.

## E    DETAILED EXPERIMENT SETTINGS

The detailed hyper-parameter settings of the experiments are listed in Table 1.

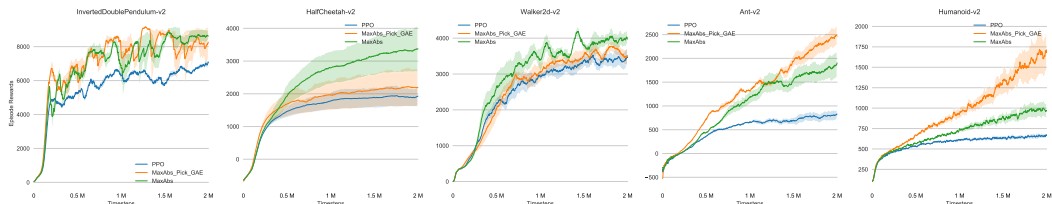

Figure 9: **Ensemble $\lambda$s.** Green: MaxAbs, Blue: PPO, Orange: MaxAbs of $\lambda$s of $\{0.1, 0.3, 0.6, 0.95\}$. Here the regulatory ratio $\rho = 0.4$. Each experiment has 5 runs.

Table 1: Detailed hyper-parameter settings of the experiments.

| Environment Type | Env Name | Hyper-Parameter Value | Alg. |
|---|---|---|---|
| Mujoco


TerrainRL | InvertedDoublePendulum-v2
HalfCheetah-v2
Walker2d-v2
Ant-v2
Humanoid-v2

Biped2D-Incline
Biped2D-Flat | elements of path ensemble= $\{1, 16, 64, 2048\}$
num_timesteps=2e6
seeds=$\{0,1,2,3,4\}$
num_envs=1
gamma=0.99
lam=0.95
optimizer=Adam
lr=lambda f: 3e-4 * f
cliprange=0.2
value_network='copy'
rho=0.4 | PPO |
| Sparse Mujoco | 2D Maze
Doom | elements of path ensemble= $\{1, 16, 64, 4000\}$
rho=0.5 | EX2 |
|  | SparseHalfCheetah | elements of path ensemble= $\{1, 16, 64, 5000\}$
rho=0.3 |  |
| Atari | PongNoFrameskip-v4

QbertNoFrameskip-v4 | minibatch=10
lengthOfTrajectory=50
rho=0.1
elements of path ensemble=$\{1, 3, 5\}$
numberOfParallelEnv=4 | A2C |
|  | SeaquestNoFrameskip-v4

BreakoutNoFrameskip-v4 | minibatch=20
lengthOfTrajectory=100
rho=0.1
elements of path ensemble=$\{1, 3, 5\}$
numberOfParallelEnv=4 |  |
|  | BeamRiderNoFrameskip-v4 | minibatch=60
lengthOfTrajectory=300
rho=0.1
elements of path ensemble=$\{1, 3, 5\}$
numberOfParallelEnv=4 |  |

