# OpenReview forum: "Regulatory Focus: Promotion and Prevention Inclinations in Policy Search"
_ICLR.cc/2020/Conference — Reject_

### Official Review · AnonReviewer2 · 2019-10-20
**Official Blind Review #2**

**Rating:** 3

**Review:**

This paper focuses on risk-aware reinforcement learning, where an agent could be encouraged to take more risk (high reward, high variance states) or avoid risk (low variance states). Risk control is instantiated by different ways of estimating the advantage of a state (max/min instead of average). Experiments on several environments show good performance of the proposed algorithm.

The paper is written clearly and the approach is straightforward. However, it's unclear what is the objective function the algorithm is optimizing by using a biased estimation of the advantage. It seems to work pretty well in practice, but I wonder how it compares to other risk-sensitive RL algorithms (e.g. those cited in the related work section).

Overall, this paper presents a simple heuristic to steer the policy towards risk-seeking / risk-avoiding directions, but could benefit from either more theoretical analysis or more empirical comparison with other methods.

**Experience Assessment:**

I have read many papers in this area.

**Review Assessment: Checking Correctness Of Derivations And Theory:**

I carefully checked the derivations and theory.

**Review Assessment: Checking Correctness Of Experiments:**

I carefully checked the experiments.

**Review Assessment: Thoroughness In Paper Reading:**

I read the paper at least twice and used my best judgement in assessing the paper.

---

### Official Review · AnonReviewer3 · 2019-10-20
**Official Blind Review #3**

**Rating:** 3

**Review:**

The paper studies the problem of advantage estimation for actor-critic RL algorithms. The key observation is that the advantage can be computed using 1-step returns, 2-step returns, etc. The paper suggests that, instead of choosing a fixed n, we should aggregate these advantageous together. If the maximum is taken, the resulting policy will be exploratory (i.e., have a "promotion focus"); if the minimum advantage is taken, the resulting policy will be risk sensitive (i.e., have a "regulatory focus").

The paper presents results on a number of tasks. A few toy examples show instances where the proposed method works better. Experiments on sparse reward environments show that taking the max advantage provides for exploration that outperforms a baseline (EX2). On a walking talk show that the min-approach can outperform state-of-the-art on-policy RL (PPO); the paper suggests this is because the min-approach is implicitly risk sensitive. Experiments on some of the Mujoco control tasks and some of the Atari games show that taking the advantage with the maximum absolute value performs well, as compared to PPO.

Overall, I think the main contribution of this paper is the finding that there exist smarter ways of computing the advantage. A second contribution is connecting the ideas of risk-sensitive and risk-seeking control with ideas from psychology (regulatory fit theory).

I am leaning towards rejecting this paper. On the one hand, the paper is well written, well motivated, and the experiments quite thorough. On the other hand, I don't think the paper is particularly useful, either from a theory or algorithmic perspective. It is not surprising that if we add an additional hyperparater to an existing RL algorithm and tune that hyperparameter, we'll do better than the existing RL algorithm. The ablation experiments in Fig 6 and Fig 7 suggest that how the advantage estimates are combined, and how they are "mixed" with a standard advantage estimate, matter a lot. Further, the fact that using a smaller number of advantage estimates worked better (point #2 on pg 5, Effect of Ensemble Size in Appendix A) suggests that the ensemble size is an important hyperparameter, and that risk-seeking / risk-aversion (i.e., regulatory vs promotion focus) cannot alone explain why the proposed method works. I think that, for this idea to be useful, it must be equipped with a fixed value for these hyperparameters that works well across a wide range of tasks (e.g., a learning rate of 3e-4 works well for Adam on most tasks), or an automated strategy for choosing this hyperparameter (e.g., the automatic entropy tuning in SAC). I would consider increasing my review if either of these were included.

A second concern is that, despite the close connections with risk-sensitive and risk-seeking control (discussed in Section 2), none of these prior works are compared against. Many of these prior works include a temperature parameter for trading off risk-seeking vs risk-aversion (e.g., \beta in Eq 11 of [Mihatsch 2002]), which is arguably a more transparent (to the user) and easier to analyze (for the RL researcher) than the order statistics used in this paper. Given the wealth of prior work on risk-sensitive and risk-seeking control, I think it's important to know whether these prior methods already solve the problem at hand (choosing between risk-seeking and risk-aversion). I would consider increasing my review if one of these methods were included as a baseline.

Minor comments
* "that humans own" -> "that humans' own"
* The max strategy seems quite closely related to UCB-based exploration. I'd be curious to learn about some discussion on the similarities/differences
* Fig 6 -- Can you add error bars to show the variance across random seeds?
* Appendix D -- How does this approach and the results shown in Fig 9 differ from standard GAE?

----------------------- UPDATE AFTER (NO) AUTHOR RESPONSE ----------------------------
The authors did not post a response, so I will maintain my vote to "weak reject" this paper. I would encourage the authors to incorporate the feedback in all reviews and submit the paper to a future conference.

**Experience Assessment:**

I have published one or two papers in this area.

**Review Assessment: Checking Correctness Of Derivations And Theory:**

I carefully checked the derivations and theory.

**Review Assessment: Checking Correctness Of Experiments:**

I carefully checked the experiments.

**Review Assessment: Thoroughness In Paper Reading:**

I read the paper at least twice and used my best judgement in assessing the paper.

---

### Official Review · AnonReviewer1 · 2019-10-24
**Official Blind Review #1**

**Rating:** 3

**Review:**

This paper presents a modification to policy gradient methods that are computed from advantage function estimates. For a given trajectory of n steps, there are n different advantage function estimates: from 1-step to n-step. GAE (Schulman, 2016) proposes to take an exponentially weighted average of these estimates to compute the policy gradients. This paper proposes instead to use order statistics to compute the policy gradient; e.g. the most optimistic estimate, the most pessimistic estimate, or the most extreme estimate.  The paper introduces a regulatory ratio: the probability of using the averaged advantage estimate vs using the order statistic, for computing the policy gradients. This hyper-parameter is justified on the optimistic case (max advantage), as a way to prevent overtly optimistic estimates. The paper conducts experiments on  different domains (sparse and dense rewards, discrete and continuous actions, fully observable and partially observable environments) which show the effect of choosing different order statistics and regulatory ratio on the policy performance.

This paper could be accepted as it presents an interesting idea with extensive experiments showing where it works and where it fails, along with some justification for the hyperparameter choices.  But there are a couple concerns about the validity of the method.

One concern is that the regulatory ratio is only justified for the max case, but not for the min or max-abs case. In addition, the choice of regulatory ratio seems to have a wildly varying impact depending on the choice of the order statistic and environment, for which we get little insights from the paper.

Another concern is that most of the results are reported only for an ensemble of 4 n-step estimators. In the appendix, the papers reports a comparison with a larger ensemble (12 n-steps estimators), which results in lower performance. This is a bit confusing: using the order statistics (max, min and max-abs), and following the reasoning presented in the paper, I would expect that increasing the ensemble size would result in better performance ( the max/min of the 4 ensemble is n upper/lower bound of the max/min of the 12 ensemble). This paper could be improved with more detailed results on the effect of the ensemble size. While the paper provides arguments for why small ensembles might suffice, it does not explain how the ensemble should be chosen, and what would happen as the ensemble size increases.

Finally, the parallels to human psychology are  bit superfluous. I understand how it might serve as an inspiration for algorithm design, but it is a bit distracting from the technical contribution of the paper.

Things to improve:

Why is max-abs missing from Figure 2?
Did you try a Rainbow-style training (changing the regulatory focus periodically over training)?

In some sentences, it looks as if insure was written in place of ensure. In general, this paper would benefit from proof reading by a proficient English speaker.

**Experience Assessment:**

I have read many papers in this area.

**Review Assessment: Checking Correctness Of Derivations And Theory:**

I assessed the sensibility of the derivations and theory.

**Review Assessment: Checking Correctness Of Experiments:**

I carefully checked the experiments.

**Review Assessment: Thoroughness In Paper Reading:**

I read the paper thoroughly.

---

### Decision · Program_Chairs · 2019-12-19

**Decision:**

Reject

**Comment:**

The authors take inspiration from regulatory fit theory and propose a new parameter for policy gradient algorithms in RL that can manage the "regulatory focus" of an agent.  They hypothesize that this can affect performance in a problem-specific way, especially when trading off between broad exploration and risk.  The reviewers expressed concerns about the usefulness of the proposed algorithm in practice and a lack of thorough empirical comparisons or theoretical results.  Unfortunately, the authors did not provide a rebuttal, so no further discussion of these issues was possible; thus, I recommend to reject.